# Envenoming by a Marine Blood Worm (*Glycera*)

**DOI:** 10.3390/toxins14070495

**Published:** 2022-07-17

**Authors:** Daniela M. Durkin, Alison N. Young, Kent Khtikian, Zuzana Karjala, Arin L. Isenstein, Bryan G. Fry, Matthew R. Lewin

**Affiliations:** 1Center for Exploration and Travel Health, California Academy of Sciences, San Francisco, CA 94118, USA; daniela@dalleradurkin.org; 2Division of Research, Ophirex, Incorporated, Corte Madera, CA 94925, USA; ayoung@calacademy.org (A.N.Y.); khtikian@kkcounsel.com (K.K.); zkarjala@yahoo.com (Z.K.); 3Department of Dermatology, Alamance Dermatology, Burlington, NC 27217, USA; arin.isenstein@gmail.com; 4Venom Evolution Laboratory, School of Biomedical Sciences, University of Queensland, St. Lucia 4072, Australia; bgfry@uq.edu.au

**Keywords:** envenoming, neurotoxicity, cytotoxicity, venom, worm, *Glyceridae*, annelid, Polychaetae, bait-diggers hand

## Abstract

Bites from venomous marine annelid ‘bloodworms’ (e.g., *Glycera* spp.) do not appear to have been described in the medical literature despite being seemingly well-known to bait diggers and fishermen. The few laboratory study reports describe their venom composition and physiological effects in vitro to be primarily proteolytic enzymes and neurotoxins apparently used for predation and defense. Herein, we present the report of a symptomatic envenoming suffered by a marine ecologist bitten while performing her field research. The local effects included a rapid onset of pain, swelling, and numbness at the bite site “as if injected with local anesthetic”. Additional signs and symptoms appearing over a two-week period were consistent with both delayed venom effects and potentially secondary infection. The late signs and symptoms resolved during a course of antibiotic treatment with doxycycline prescribed as a precaution and lack of resources to consider a wound culture. Comments about annelid bites sporadically appear in the popular literature, especially pertaining to the fishing industry, under names such as ‘bait-diggers hand’. While these bites are not known to be dangerously venomous, they seem to produce painful local symptoms and possibly increase the risk of marine bacterial infections that could be associated with more serious outcomes. More cases need to be formally described to better understand the natural history of these types of envenomation.

## 1. Introduction

The case described herein is the bite from a bloodworm (*Glycera*) delivered to the non-dominant hand of a marine biologist while surveying invertebrate populations in tidal flats typically inhabited by these predatory annelid worms. The patient, an otherwise healthy female 40-year-old field biologist, collected and was bit by an annelid (worm) from the genus *Glycera* during a survey of tidal pools and mud flats just south of San Francisco. In transferring the worm, it bit her on the volar surface of the middle phalanx of her left second (index) digit and she ‘flung it off’, with the resulting puncture wounds evident in the immediate aftermath (Figure 1, day 0). She described the bite as feeling much like a ‘bee sting’ with pain, itching, and local anesthesia at the bite site for about a day, after which the puncture wounds closed and erythematous macules developed in their place (Figure 1, day 1). On day 8, the patient sought medical consultation from the institution’s expedition physician as the affected digit was edematous and presented surprising discoloration and by the following morning was ‘very swollen, warm but not painful’ or tender but had pronounced vesicles at the puncture sites. The patient sought consultation due to the unexpected signs and symptoms from the bite as she was in transit for long-distance travel that would be worrisome if care was delayed (Figure 1, day 8). There were no medical facilities capable of attempting culture and empiric antibiotics were prescribed as a precaution despite the unclear etiology of the signs and symptoms.

A delayed envenoming reaction, infection, and inflammatory response to a retained foreign body (e.g., jaw fragment) were considered. There were no signs or symptoms of a retained foreign body since the bite and imaging was not deemed urgent at the time of consultation, though this would have been appropriate (e.g., x-ray, ultrasound) had there been suspicion at any time. Despite the lack of tenderness, lymphangitis, or lymphadenopathy, a secondary bacterial infection was still considered possible, if not likely. Doxycycline was administered on day 8 at 100 mg orally twice a day for seven days to cover known marine infections such as those caused by *Mycobacterium*, *Vibrio, Aeromonas, Shewanella*, and others [1,2,3]. Doxycycline was tolerated well, and sun precautions were taken during its treatment to avoid phototoxic reactions common with tetracycline antibiotics. By the following day, there was a significantly lesser degree of surrounding purpura than that seen on day 8 (Figure 1, day 9).

By day 11 (day 3 of antibiotics), a decrease in edema and erythema was noted while the vesicles increased in prominence. By day 12 following the bite (day 4 of antibiotics), the vesicles began to dry out, becoming hyperkeratotic (Figure 1, day 12) and the erythema and edema so striking on day 8 had been largely resolved. The papules began to desquamate and resolve between days 15 and 19. There was no residual tenderness, and a full range of motion returned, as reported by the patient. There was no follow-up until the following year. The patient never experienced fevers, chills, nausea, vomiting, diarrhea, chest pain, shortness of breath, nor myalgias. She did not have signs or symptoms of lymphatic streaking or tender lymphadenopathy at any point following the envenomation.

## 2. Discussion

Marine worms within the *Glycera* genus are often sold in bait fishing shops as ‘bloodworms’, a term that broadly describes numerous hemoglobin expressing invertebrates including polychaete worms in the family *Glyceridae* and even worm-like insect larvae such as those in the Chironomidae family found in marine, brackish, and freshwater environments. Glycerids themselves (*Glycera* spp.) are venomous predatory worms possessing four jaws fused with hollow fangs comprising an unusual composition of equal parts protein, melanin, and ~10% elemental copper, which complexes with melanin to produce a remarkably durable and wear-resistant venom apparatus with curved fangs [4]. The jaws are housed far from the oral opening at the end of a tubular proboscis, externalized anteriorly for predation and defense while each is connected to a venom gland. Upon biting down, the venom glands empty through ducts at the roots of the jaws to immobilize and digest prey or repel predators and competitors (Figure 2).

While other marine invertebrate venoms such as those from cnidarians (e.g., Cubozoa) and mollusks (e.g., Conidae) are very well described, the venom composition of *Glyceridae* has not been extensively studied. The venom glands of the studied species, *G. dibranchiata*, *G. fallax*, and *G. tridactyla*, have been found to contain mixtures of large, globular proteins and disulfide-bond, cross-linked peptides with toxic effects in vitro [5]. Some of these toxins represent convergent evolution using the same protein frameworks, such as ones that use the same SPRY-framework as the pain-inducing stonustoxin from the stonefish species [6] (*Synanceia* spp.). In addition, *G. convoluta* was found to contain a protein-toxin (“glycerotoxin”) capable of stimulating neurosecretion [7]. Research into the venoms of other marine worms has also been limited, but a diversity of toxins has been obtained from the species that have been examined, ranging from cytotoxins to neurotoxins [8,9]. A recent paper on the venom of the ribbon worm *Antarctonemertes valida* [10] revealed distinct defensive and predatory toxins, a scenario analogous to the separate defensive and predatory venoms produced by some cone snails (*Conus* spp.) [11,12].

Glycerid bites may be a recognized occupational hazard amongst those in the bait fishing industry and have sometimes been mentioned in popular media such as magazines and even videos. However, to our knowledge, this is the first report in the medical literature describing the occurrence and natural history of a bite from a venomous polychaete in the genus *Glycera*. Even as the proteomic and toxinological features of venom for several genera, including *Glycera*, have been described, the clinical features of bites from venomous worms in this or related annelid genera have not appeared in the peer-reviewed literature. Larger species of *Glycera* worms are commonly used as bait for salt and brackish-water fishing, and Maine has long been a center of the Glycerid worm-harvesting industry [13,14].

As observed in this patient’s case, fluid filled vesicles formed as the result of inflammation in the skin, which was not clearly defined as originating from either infection or delayed venom effects (Figure 1, day 9). When either vesicles (as in this case) or pustules (e.g., following *Solenopsis* “fire-ant” bites) dry, the stretched skin overlying the fluid-filled papules become hyperkeratotic, (Figure 1, day 12) and then peels off (Figure 1, days 15 through 19). The vesicles observed in this case are not unsimilar in appearance to those seen in eczematous processes such as allergic contact dermatitis (e.g., poison ivy), infectious processes (e.g., herpes simplex virus, varicella-zoster virus, and bullous impetigo), autoimmune blistering diseases (e.g., bullous pemphigoid, pemphigus vulgaris, and dermatitis herpetiformis), or insect bite hypersensitivity reactions. Pustules arise in pustular psoriasis, infectious processes (e.g., herpes simplex virus, candidiasis, and folliculitis) and *Solenopsis* bites, which can produce an extreme itch and sterile pustules after the acutely painful venom effects have worn off. All of these processes were considered, though none were deemed likely.

While immediate venom effects may be clear, a comprehensive understading of the natural history of *Glycera* bites and risk of secondary infection remains unknown even in light of this initial case report. Additionally, microbes can be introduced through a bite wound such that the intersection of venom effects and microbial intrusion are not necessarily obvious from a single case. The speed of resolution from day 8 to day 9, for example, is not clearly related to antibiotic therapy, and delayed systemic and local venom effects are known for multiple venoms, including those from jellyfish, hymenoptera stings, and snakebites in varying degrees as the result of direct but delayed toxicity or from delayed inflammatory reactions to the venoms themselves [15,16,17].

Marine microbes can cause significant infections with indolent or rapid and even lethal progression, but apart from the delayed episode of discoloration, the wound in this case was not clearly infected [1,2,3]. Long-term deformities such as seal finger, a condition even described by the Vikings as *spekkflegmonnen* or ‘blubber finger’ and believed to be caused by marine *Ureaplasma* or related organisms likely entering through small cuts on the hands of seal hunters, have recently been reported as an occupational hazard for marine biologists [18]. Tetracycline-based antibiotics are often reported to be effective against marine organisms otherwise unresponsive to penicillin and cephalosporin antibiotics. *Streptococcus* or *Staphyloccus* were considered unlikely in this case due to the absence of pain, lymphadenopathy, purulence, or any constitutional signs and symptoms [1,2,3].

Our estimation is that there was a combination of immediate venom effects and possibly an infection, although the signs and symptoms were not obviously consistent with infection. Antibiotic treatment was empirically based as a precaution and on the probability of the success of the selected antibiotic in the event of an infection. As an initial formal case description of *Glycera* envenomation, more cases should be described in order to further understand the natural history of these types of envenomation and their appropriate management with or without antibiotics.

## Figures and Tables

**Figure 1 toxins-14-00495-f001:**
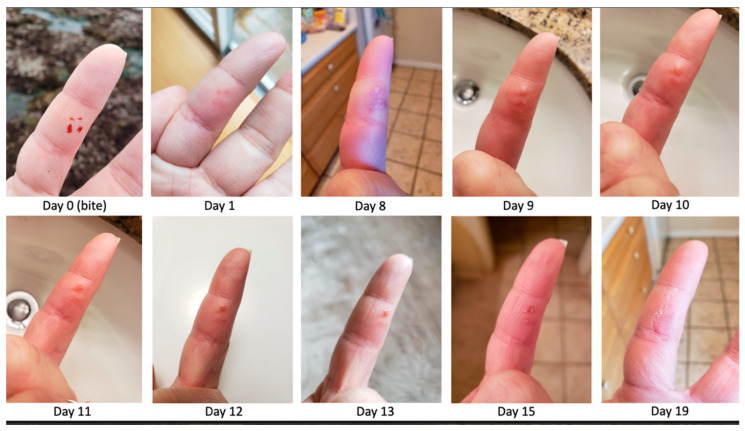
The natural history of *Glycera* bites is unknown. Time course of the bite with an acute envenoming syndrome (day 0 and day 1) followed by possible delayed venom reaction, infection, or combination of both (day 8) and resolution (e.g., days 9 through 19).

**Figure 2 toxins-14-00495-f002:**
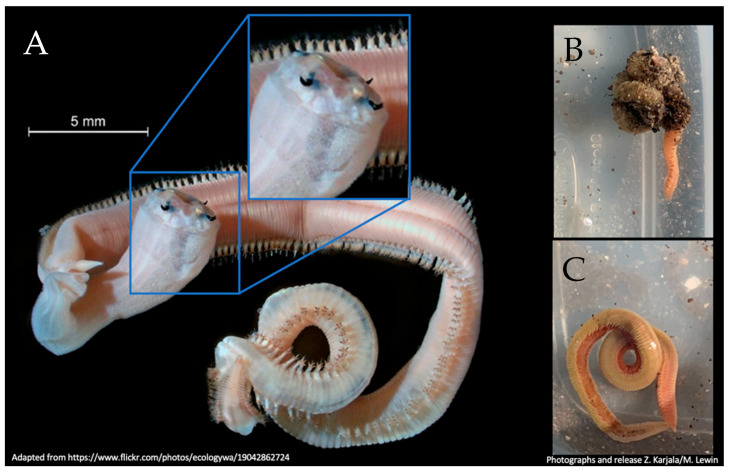
(**A**) Characteristic four-fanged jaws of *Glycera americana* typical of the *Glycera* genus. Fangs reside inside the abdominal cavity and can be rapidly externalized for predation or defense when threatened such as what might have occurred in the case of this patient handling what appears to be an innocuous looking annelid, left top and bottom. (**B**,**C**) *Glycera* spp. worm similar to that which bit the patient was collected, photographed, and released at the locality where the patient was bitten.

## Data Availability

Not applicable.

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
