# Peer review of "Envenoming by a Marine Blood Worm (*Glycera*)"

_toxins, 2022, doi:10.3390/toxins14070495_

Round 1

Reviewer 1 Report

This is an interesting Index Case Report of Envenoming by a marine blood worm 2 (Glyceridae). This is the first report of symptomatic envenoming suffered by a marine ecologist bitten during the performance of her field research. Local effects and clinical course have been carefully described and supported by iconographic documentation, hypothesizing delayed venom effects and/or secondary infection treated with doxycycline orally. Comments about annelid bites sporadically appear in popular literature, especially pertaining to the fishing industry. These bites may produce painful local symptoms and secondary infections associated with more serious outcomes. In particular, the latter is observed with marine Vibrio and mycobacterial infection, requiring emergency medical care.

Minor changes are indicated in the text.

Author Response

Reviewer #1

 Emergency

Agreed, change made.

Please check last two references Multiple references have been added. The two referenced by the reviewer have been checked and confirmed (now 17&18).

Reviewer 2 Report

REVIEW

4th line: spp. is not written in italicized characters. See the zoological nomenclature, please.

7th line: Claiming to be publishing the first case of an accident is virtually impossible these days. The authors should remove this sentence and let future citations about the article decide whether or not this is the first description...

11th line: For the case report, it is essential to differentiate between delayed venom effects and/or secondary infection. One aspect speaks of late action of the venom, something difficult to prove and absent from publications, even the experimental ones. Secondary infection would already be more likely, but the sentence form cannot remain as it currently is.

17th line: Zoological families is not written in italicized characters...

18th line: “bait-diggers hand” is a keyword: There are rules to cite Keywords.

28th line: Glyceridae is not a genus.

44th line: The authors' hypotheses are valid, but it seems clear that retention of foreign bodies is not an option, given the shape and size of the fangs. I don't think there would be any breakage when bitten. Additionally, flare-ups of inflammation in sting areas after the initial phase invariably indicate secondary infections, usually bacterial. The authors should remember that bacterial infections in marine environments even occur through seawater in open wounds...

48th line: Atypical mycobacteriosis and Vibrio vulnificus infections do not remember the patient's signs and symptoms. Mycobacteria are slow to install and have an ascending lymphangitis pattern. Vibrio vulnificus causes an explosive infection, with cellulitis and deep tissue necrosis. Authors should research other more likely causes before citing these agents.

53th line: Edema is a long-standing presence in accidents caused by fish of the Synanceiidae family, whose venom has certain similarities with the components of marine worm toxins. And the regression cannot be accelerated, only with physical therapy and time...

57th line: What do the authors understand by “complete recovery”? What remained for a year at the accident site?

61-67th lines: The discussion should not quote what has already been said in the Introduction, but be restricted to the case described.

61th line: Glyceridade is not a genus...

63th line: Chironomidae is not written in italicized characters.

72th line: The Figure 2 should be in the Introduction.

78th line: Conidae is a zoological Family.

104th line: “Bumps”? Eruption, right?

104th line; Vesicles are always vesicles and pustules already appear as structures filled with purulent secretion (polymorphonuclears). It may be sterile. There were never pustules in this case, the old vesicles, after a few days, become cloudy and confuse the non-specialist.

106th line: The vesicles are different in what aspects from other cited dermatoses? The citation pure and simple does not clarify the reader in a future differentiation. It is necessary that the authors complement the information.

111th line: As the pustules, by definition, already appear with purulent content, I do not think that the discussion about them is appropriate for the case report. Solenopsis  stings really cause pustules, but herpes vírus provokes vesicles.

115th line: Glycera: italic.

122th line: “Blubber finger” (Erysipelothrix rhusiopathiae), atypical mycobacterioses, Vibrio vulnificus and Shewanella are possible agentes of marine infections, but they are not so common. The authors shoud Search the most probable causes of secondary infections after wounds in marine environments and modify the text.

132th line: Did the authors not culture the wounds for bacteria? This would be critical for differentiating between the effects of the venom and secondary infection.

Author Response

Reviewer #2

4th line: spp. is not written in italicized characters. See the zoological nomenclature, please.

We thank the reviewer for this check and agree that for non-microbial names, Family and above remain in Roman type.

7th line: Claiming to be publishing the first case of an accident is virtually impossible these days. The authors should remove this sentence and let future citations about the article decide whether or not this is the first description...

We appreciate the point, but it is exactly this finding from extensive literature search that strongly suggests this is the first formally reported case in the medical literature. Nevertheless, we have made the language more circumspect.

11th line: For the case report, it is essential to differentiate between delayed venom effects and/or secondary infection. One aspect speaks of late action of the venom, something difficult to prove and absent from publications, even the experimental ones. Secondary infection would already be more likely, but the sentence form cannot remain as it currently is.

 We appreciate this point. Delayed venom effects are quite well described from other envenomings such as snakebite. Arguing against infection is that the finger was neither painful nor tender. This suggests a local coagulopathy or cytotoxic injury similar to those seen in other envenoming syndromes. She clearly had early envenoming effects (described as her immediate signs/symptoms). We agree with the reviewer that it is not clear if the later discoloration was the result of primary or secondary venom effects or infection resulting from tissue compromise.

We have made these elements clearer and thank the reviewer for this point of discussion.

17th line: Zoological families is not written in italicized characters...

Agreed, change made.

18th line: “bait-diggers hand” is a keyword: There are rules to cite Keywords. 

Noted. Agree.

28th line: Glyceridae is not a genus.

Agreed. Thank you. We have corrected “Glyceridae” to “Glycera”.

44th line: The authors' hypotheses are valid, but it seems clear that retention of foreign bodies is not an option, given the shape and size of the fangs. I don't think there would be any breakage when bitten.

We appreciate you bringing this topic to our attention. In the paper, we did in fact agree with the conjecture you made above. The treating physician (MRL) considered the retention of a foreign body, but it was considered unlikely due to the symptoms and signs of the patient and pertinent negatives such as absence of foreign body retention (now added to the text as a pertinent negative)

We have made these points clearer and thank the reviewer for this point of discussion.

 Additionally, flare-ups of inflammation in sting areas after the initial phase invariably indicate secondary infections, usually bacterial. The authors should remember that bacterial infections in marine environments even occur through seawater in open wounds...

We have clarified this point in the text and appreciate the point. We respectfully disagree that inflammatory flare-ups following envenoming by invertebrates is invariably the result of infection and have weighted our revised discussion accordingly.

48th line: Atypical mycobacteriosis and Vibrio vulnificus infections do not remember the patient's signs and symptoms. Mycobacteria are slow to install and have an ascending lymphangitis pattern. Vibrio vulnificus causes an explosive infection, with cellulitis and deep tissue necrosis. Authors should research other more likely causes before citing these agents.

We have broadened the differential diagnosis of infective agents accordingly. We agree that the atypical mycobacterial infections seen tend to be more granulomatous in appearance, typically presenting with a solitary erythematous nodule or a few papules/nodules that move proximally along lymphatic pathways. These infections are not typically vesicular, edematous, or cellulitic.

53rd line: Edema is a long-standing presence in accidents caused by fish of the Synanceiidae family, whose venom has certain similarities with the components of marine worm toxins. And the regression cannot be accelerated, only with physical therapy and time...

Thank you for bringing to our attention. This point similarly considerably balances the probability that the effects are as likely to have been the result of an envenoming reaction as those resulting from an infection. We agree that there is a risk/benefit of treatment as it pertains to regression or worsening, and it was this consideration that resulted in the decision to treat empirically with doxycycline. The finger’s discoloration was largely cleared by antibiotic day 4, but this could also result from clearing of cellular breakdown products and hemoglobin breakdown products. Neither is ruled out (infection or delayed venom effects).

57th line: What do the authors understand by “complete recovery”? What remained for a year at the accident site?

Thank you for pointing this out. We have clarified “complete recovery” by including that the finger had returned to its normal state prior to the bite.

61-67th lines: The discussion should not quote what has already been said in the Introduction, but be restricted to the case described.

Noted.  

61st line: Glyceridade is not a genus...

Agreed, changed “Glyceridae” to Glycera.

63rd line: Chironomidae is not written in italicized characters.

Agreed, “Chironomidae” is a family and therefore is not italicized.

72nd line: The Figure 2 should be in the Introduction.

 Noted.

78th line: Conidae is a zoological Family.

Agreed, “Conidae” has been italicized.

104th line: “Bumps”? Eruption, right?

Thank you for pointing out possible confusion here. We have changed “bumps” to “papules” to be more accurate with the description.

104th line; Vesicles are always vesicles and pustules already appear as structures filled with purulent secretion (polymorphonuclears). It may be sterile. There were never pustules in this case, the old vesicles, after a few days, become cloudy and confuse the non-specialist.

Agreed, there were not any pustules in this case, as we alluded to. It was stated that the vesicles which were present were commonly confused for pustules, not that they were in fact pustules. We recognize that the discussion of pustules is not helpful nor relevant in this case and it has been reduced.

106th line: The vesicles are different in what aspects from other cited dermatoses? The citation pure and simple does not clarify the reader in a future differentiation. It is necessary that the authors complement the information.

We agree. The vesicles in this case appear to be  located at the sites of the puncture wounds. The vesicles with HSV tend to be grouped unlike in this case. Autoimmune blistering diseases have larger lesions, bullae. Bullous insect bites present as a solitary vesicle or bulla at the site of an insect bite. They are pruritic and represent a delayed hypersensitivity reaction in sensitized individuals. The vesicles and bulla in allergic contact dermatitis also represent a delayed hypersentivity reaction and are often in geometric arrangements at sites of exposure.Fire ant bites create vesicles and pustules within 24 hours at the bite site due to direct effects of the venom. The timing of the onset of vesicles in this case is delayed beyond what is normally seen in hypersensitivity reactions, 8 days in this case vs 2-3 days in typical hypersensitivity reacitons. The edema, discoloration, and onset at day 8 point towards an infectious etiology or perhaps a delayed effect of a toxin.

111th line: As the pustules, by definition, already appear with purulent content, I do not think that the discussion about them is appropriate for the case report. Solenopsis  stings really cause pustules, but herpes virus provokes vesicles.

Agree. See above comment.

115th line: Glycera: italic.

Agreed, “Glycera” has been italicized 

122nd line: “Blubber finger” (Erysipelothrix rhusiopathiae), atypical mycobacterioses, Vibrio vulnificus and Shewanella are possible agentes of marine infections, but they are not so common. The authors should Search the most probable causes of secondary infections after wounds in marine environments and modify the text. 

Agree. Per our response above, we have expanded the differential diagnosis of marine and terrestrial organisms that might cause this type of reaction. We again thank the reviewer for expanding the utility of the manuscript.

132nd line: Did the authors not culture the wounds for bacteria? This would be critical for differentiating between the effects of the venom and secondary infection.

We did not, as the patient was in transit and treatment was empirical based on the probability of success of the selected antibiotic. The patient was in need of assistance with incomplete information and resources as happens sometimes during travel and especially in wilderness settings. It would have been reasonable to consider double coverage with a cephalosporin, too (now reflected in the text). There was no time nor resources to culture the wound.  However, we do agree that the process would be revealing in determining the causality of the various effects

Reviewer 3 Report

This is a very interesting first case report of envenomation following a marine blood worm bite. The authors have written the case elegantly without any major issues. I would suggest the authors to address the following comments in the revised manuscript to help the readers to understand this case better in a wider context.

1. Please include a brief introduction to set the context prior to describing the case. 

2. Did the authors obtain a written consent from the patient to publish this data? Please confirm this at the beginning of case description or at the end. 

3. Please highlight the gender of the patient and who confirmed the identity of the worm. 

4. It would be very helpful to include all laboratory and other relevant investigations performed on this patient. Did they perform microbial culture to identify the bacteria at affected site? Any CT or MRI performed to rule out the possibilities of the presence of a foreign body at the bite site? 

5. Please mention any other medications used in this patient in addition to antibiotics. 

6. Since this venom contains proteolytic enzymes and neurotoxins, are they likely to induce any bleeding or clotting or neurological complications? did the patient experience any other symptoms other than the local symptoms reported? Any long term effects (minor or major) over one year? It would be very helpful if the authors could cover all these aspects in the case description. 

7. In the discussion, please write few sentences about the nature of toxins reported in this venom and the type of envenomation effects they are expected to induce. What would you advise to clinicians who normally treat this type of cases? 

Author Response

Reviewer #3

is is a very interesting first case report of envenomation following a marine blood worm bite. The authors have written the case elegantly without any major issues. I would suggest the authors to address the following comments in the revised manuscript to help the readers to understand this case better in a wider context.

  1. Please include a brief introduction to set the context prior to describing the case. 

Noted. Thank you for this suggestion. We have made this change.

  1. Did the authors obtain a written consent from the patient to publish this data? Please confirm this at the beginning of case description or at the end. 

Yes. This was included in the paper upon submission (under section of disclosures and acknowledgments. We will make sure it is present in proof to ensure this information is retained.

  1. Please highlight the gender of the patient and who confirmed the identity of the worm. 

Agree. Briefly, the patient is a research biologist, and the identity of the worm was known to her and other scientific staff present as within their scope of expertise is marine invertebrate biology.

  1. It would be very helpful to include all laboratory and other relevant investigations performed on this patient. Did they perform microbial culture to identify the bacteria at affected site? Any CT or MRI performed to rule out the possibilities of the presence of a foreign body at the bite site? 

Treatment was empirical and the patient had no foreign body sensation. Had the signs/symptoms not cleared, this would have been indicated. We have added this to the discussion as it pertains to basic and follow-up care as well as being generally educational. To that point: CT, MRI, culture, not performed, but we have added possibilities for imaging that might be done (e.g., cross sectional imaging by radiograph or ultrasound)

  1. Please mention any other medications used in this patient in addition to antibiotics. 

Thank you for reminding us of this pertinent negative. No concomitant medications were being taken.  

We have also included clarifying comments on the patient’s allergies to medications, history of vaccinations, etc., and the fact that she was up to date on Tetanus shot.

  1. Since this venom contains proteolytic enzymes and neurotoxins, are they likely to induce any bleeding or clotting or neurological complications? did the patient experience any other symptoms other than the local symptoms reported? Any long-term effects (minor or major) over one year? It would be very helpful if the authors could cover all these aspects in the case description. 

We have clarified this in the text; thank you for making this point.

  1. In the discussion, please write few sentences about the nature of toxins reported in this venom and the type of envenomation effects they are expected to induce. What would you advise to clinicians who normally treat this type of cases? 

We have made this element of the discussion more prominent. 

Round 2

Reviewer 2 Report

TITLE

Case Report of Envenoming by a marine blood worm (Glycera genus)

ABSTRACT (suggestions to be included also in thet text)

7th Line: The Authors should understand that whoever says whether a case report is the first described are the next publications that cite it. With the advances of electronic publications, it is no longer possible to use phrases like this. I have already advised them to change the text in the previous revision, in order to improve the report. It must be said that no cases caused by marine worms were found in the bibliographic research, citing at least two important sources of research. Please, change the text.

11th line: Antibiotics do not treat envenomations... The phrase that says "delayed effects of poison OR secondary infection treated with doxycycline" makes no sense. If there was remission with antibiotics, it is because there was an infection. If there was only effects of the envenomation, the regression was spontaneous and not caused by the antibiotic.

16th line: There is no necessity to cite dangerous, but rare agentes of bacterial infections, since no Vibrio or mycobaterial were demonstred in annelid´s bites. I think that the authors should  only say “bacterial infections”. An extensive discussion of infections in the aquatic environment does not fit in the manuscript, since the infection was not proven. What was the reason for not having performed the culture for bacteria and antibiogram? Despite the justification of the case being published due to its rarity, it is a fact that will be questioned when it is published and I would like to have an answer before approval.

Keywords: Glyceridae is a Family zoological name and it is not graphed in italic characters.

Key contribution: a “rare case” not the “first case”.

Author Response

We appreciate the opportunity to refine the manuscript a second time. We have greatly reduced the discussion of specific marine microbes while retaining the discussion of clinical decision-making.  The reviewer’s perspective is appreciated and we have made as many compromises as we think appropriate. At this point, I respectfully ask for a decision so that we can decide if we wish to withdraw the submission and submit elsewhere.  This is a very unusual case and we think we have characterized it and the thinking behind the management quite clearly. Given that it was a field medicine case and not a hospital or clinic case, it is unreasonable to expect we would attempt to culture the wound. Emergency physicians are trained to make decisions with incomplete information. That was the milieu under which the risk-benefit analysis of clinical management was made.  Whether one agrees with the use of antibiotics or not (or the choice of antibiotics) it is reported faithfully and it does not alter the unique nature of the case.

We thank the reviewer for the passionate and helpful positions from which we have made revisions we think are in the spirit of the review and true to our interpretation of the case.

To that end, we request an immediate decision from the Editor.

Respectfully,

Matthew R. Lewin, MD, PhD—Corresponding Author (and treating expedition physician)

Specific Responses:

7th line: The Authors should understand that whoever says whether a case report is the first described are the next publications that cite it. With the advances of electronic publications, it is no longer possible to use phrases like this. I have already advised them to change the text in the previous revision, to improve the report. It must be said that no cases caused by marine worms were found in the bibliographic research, citing at least two important sources of research. Please, change the text.

Thank you for the suggestion to make a more careful statement. We have been even more circumspect on this point. At this point, we stand by our interpretation of the literature and our description of the case well within traditional bound. We believe we have made a very reasonable compromise on this and respectfully suggest it should now be left to the Decision Editor if the reviewer disagrees with our edits on this point.

11th line: Antibiotics do not treat envenomations... The phrase that says "delayed effects of poison OR secondary infection treated with doxycycline" makes no sense. If there was remission with antibiotics, it is because there was an infection. If there were only effects of the envenomation, the regression was spontaneous and not caused by the antibiotic.

Thank you for bringing the confusion of the wording of this sentence to our attention. We have edited the line to better reflect the meaning we intend to convey.

16th line: There is no necessity to cite dangerous, but rare agents of bacterial infections, since no Vibrio or mycobacterial were demonstrated in annelid´s bites. I think that the authors should only say “bacterial infections”. An extensive discussion of infections in the aquatic environment does not fit in the manuscript since the infection was not proven. What was the reason for not having performed the culture for bacteria and antibiogram? Despite the justification of the case being published due to its rarity, it is a fact that will be questioned when it is published, and I would like to have an answer before approval.

As we have stated in the text: We did not perform a culture as the patient was in transit and treatment was empirical based on the probability of success of the selected antibiotic. The patient needed assistance with incomplete information and resources as happens sometimes during travel and especially in wilderness settings. It would have been reasonable to consider double coverage with a cephalosporin, too (now reflected in the text). There was simply no time nor resources to culture the wound. However, we do agree that the process would be potentially revealing in determining the causality of the various effects observed.  This was not a hospital or clinic case. It was a field medicine case.